# A Mating Procedure for Genetic Transfer of Integrative and Conjugative Elements (ICEs) of Streptococci and Enterococci

**DOI:** 10.3390/mps4030059

**Published:** 2021-08-28

**Authors:** Francesco Iannelli, Francesco Santoro, Valeria Fox, Gianni Pozzi

**Affiliations:** Laboratory of Molecular Microbiology and Biotechnology, Department of Medical Biotechnologies, University of Siena, 53100 Siena, Italy; santorof@unisi.it (F.S.); valeria.fox@student.unisi.it (V.F.); gianni.pozzi@unisi.it (G.P.)

**Keywords:** horizontal gene transfer, conjugation, MGE, ICE, conjugative transposon, streptococci, enterococci

## Abstract

DNA sequencing of whole bacterial genomes has revealed that the entire set of mobile genes (mobilome) represents as much as 25% of the bacterial genome. Despite the huge availability of sequence data, the functional analysis of the mobile genetic elements (MGEs) is rarely reported. Therefore, established laboratory protocols are needed to investigate the biology of this important part of the bacterial genome. Conjugation is a mechanism of horizontal gene transfer which allows the exchange of MGEs among strains of the same or different bacterial species. In streptococci and enterococci, integrative and conjugative elements (ICEs) represent a large part of the mobilome. Here, we describe an efficient and easy-to-perform plate mating protocol for in vitro conjugative transfer of ICEs in streptococci (*Streptococcus pneumoniae*, *Streptococcus agalactiae*, *Streptococcus gordonii*, *Streptococcus pyogenes*), *Enterococcus faecalis,* and *Bacillus subtilis*. Conjugative transfer is carried out on solid media and selection of transconjugants is performed with a multilayer plating. This protocol allows the transfer of large genetic elements with a size up to 81 kb, and a transfer frequency up to 6.7 × 10^−3^ transconjugants/donor cells.

## 1. Introduction

The three major mechanisms of horizontal gene transfer in bacteria are conjugation, transformation, and transduction. Mobile genetic elements (MGEs), including conjugative and integrative elements (ICEs) and prophages, shape the bacterial genome and are responsible for genome evolution [1]. Conjugation enables the genetic exchange of MGEs, which provide a major contribution to the spread of antimicrobial resistance and virulence, by recruiting new resistance and virulence genes and facilitating their dissemination [2]. Genome-wide DNA sequencing disclosed the presence of a large number of uncharacterized MGEs, whose open reading frames are often automatically annotated as conserved genes of unknown function [3]. In fact, the nature of the mobile elements and their transfer mechanisms have been clarified only in few cases [4,5,6,7]. ICEs account for the majority of streptococcal and enterococcal MGEs [8]. To elucidate transfer mechanisms and their regulation it is essential to develop an established protocol for efficient conjugal transfer of ICEs also from encapsulated clinical bacterial strains. In this work, we developed a successful plate mating protocol for in vitro transfer of large ICEs with a size up to 81 kb in streptococci (*Streptococcus pneumoniae*, *Streptococcus agalactiae*, *Streptococcus gordonii*, *Streptococcus pyogenes*), *Enterococcus faecalis,* and *Bacillus subtilis*. 

## 2. Materials and Equipment

Deionized H_2_O;Tryptic soy broth (TSB), agar technical (BD, Difco, USA);Defibrinated horse blood (Liofilchem, Italy);Antibiotics, glycerol, ethanol (Sigma-Aldrich, USA);Filter 0.2 µm (Minisart, Sartorius, Germany);Petri dishes, tubes, serological pipets, microtubes, micropipette tips, cotton swabs, toothpicks, syringes (Sarstedt, Germany);Laboratory glassware (Schott, UK);Incubators (KW Apparecchi Scientifici, Italy);Spectronic GENESYS 200 spectrophotometer (Thermo Scientific, USA);Heat block (FALC Instruments, Italy);White light transilluminator;VAPOUR-Line autoclave (VWR Avantor, USA).

## 3. Methods

A schematic diagram of the plate mating protocol is reported in Figure 1.

### 3.1. Preparation of Media

Liquid medium: Dissolve 30 g of TSB dehydrated medium in 1 L of deionized H_2_O and autoclave at 121 °C for 15 min (see Note 1). Add the appropriate antibiotics at the following concentrations when required: 3 µg mL^−1^ chloramphenicol, 0.5 µg mL^−1^ erythromycin, 500 µg mL^−1^ kanamycin, 10 µg mL^−1^ novobiocin, 100 µg mL^−1^ spectinomycin, 500 µg mL^−1^ streptomycin, 5 µg mL^−1^ tetracycline, 25 µg mL^−1^ fusidic acid, 25 µg mL^−1^ rifampicin.Solid medium: Add 1.5% agar to TSB liquid medium and autoclave at 121 °C for 15 min. Equilibrate TSA (TSB and agar) at 48 °C for 20 min. Add 5% defibrinated horse blood and the appropriate antibiotics at the concentrations reported in step 1 when required. Mix and pour 25 mL in each Petri dish. Leave to solidify for 20 min, dry at 42 °C for 30 min and store at 4 °C.Antibiotics such as kanamycin, novobiocin, spectinomycin, or streptomycin are resuspended in deionized H_2_O, sterilized by filtration with a 0.2 µm filter; chloramphenicol, erythromycin, tetracycline, and fusidic acid are resuspended in absolute ethanol; rifampicin is resuspended in methanol. Stock solutions are as follows: 10 mg mL^−1^ chloramphenicol, 2.5 mg mL^−1^ erythromycin, 100 mg mL^−1^ kanamycin, 10 mg mL^−1^ novobiocin, 50 mg mL^−1^ spectinomycin, 100 mg mL^−1^ streptomycin, 5 mg mL^−1^ tetracycline, 5 mg mL^−1^ fusidic acid, 25 mg mL^−1^ rifampicin. Antibiotics are stored in 1 mL aliquots at −20 °C (see Note 2).Glycerol is diluted two-fold (50%) in TSB.

### 3.2. Pre-Mating Preparation of Cells

Thaw frozen starter cultures at 37 °C.Pre-warm TSB at 37 °C.Dilute frozen cultures 100 fold in TSB containing antibiotics and incubate at 37 °C.Grow donor and recipient cells separately until late exponential phase.Record the OD_590_ on a semi-log paper.Draw the growth curve and determine the duplication time.Freeze 2 mL mating starter cells at OD_590_ = 0.8 (approximately 5 × 10^8^ CFU mL^−1^) in 10% glycerol (see Note 3).

### 3.3. Plate Mating

Thaw frozen mating starter cultures (see Note 4).Mix 1:10 donor cells (100 µL) and recipient cells (900 µL) in a 1.5 mL microtube (see Note 5).Centrifuge the mixed cells at room temperature for 15 min at 3000 × g.Discard supernatant and resuspend the pelleted cells in 0.1 mL of TSB.Plate mixture on a blood-agar plate and incubate at 37 °C for 4 hours in a 5% CO_2_ enriched atmosphere.Harvest cells by scraping from the plate with a cotton swab and dissolve in a 1 mL of TSB/Glycerol 10%.Freeze mating samples at −70 °C (see Note 6).

### 3.4. Multilayer Plating

Prepare TSA medium and equilibrate at 48 °C for 20 min.Pour a 17 mL base layer of TSA in Petri dishes and let the medium solidify.Dispense 2 mL of TSB supplemented with 10% horse blood in 5 mL slip-cap tubes.Put 13 mL slip-cap tubes into a heat block at 48 °C and distribute 6 ml of TSA per tube.Add appropriately diluted mating reactions into the 2 mL-TSB-containing tube (see Note 7).Combine 6 ml of TSA with the 2 mL of TSB blood cells, shake, and pour onto plate.Let the medium solidify, incubate at 37 °C for 90 min (phenotypic expression).Add an 8 ml third layer of TSA containing the appropriate antibiotics (see Note 8).Let the medium solidify, then incubate at 37 °C overnight (see Note 9).Score transconjugant cells, calculate conjugation efficiency as transconjugant per donor cells.

### 3.5. Genetic Analysis of Transconjugants

Fit blood-agar plates on grids of the transilluminator device (see Note 10).Pick single colony isolates from conjugation plates by using sterile toothpicks and transfer to the plates placed on the illuminated grid.Incubate at 37 °C overnight in a 5% CO_2_ enriched atmosphere.Check the phenotypes of transconjugants (see Note 11).Isolate transconjugants from the genetic analysis plates on new plates containing the appropriate antibiotic.Incubate at 37 °C overnight in a 5% CO_2_ enriched atmosphere.Grow single colony isolates in TSB containing the appropriate antibiotic.Freeze transconjugant starter cultures (in exponential phase, OD_590_ = 0.2–0.3) in 10% glycerol at −70 °C.

## 4. Notes

Other media, such as brain heart infusion (BHI) (Oxoid, UK) for streptococci and enterococci can be used. For *Bacillus subtilis* the Luria–Bertani (LB) medium (BD, Difco) was used.Due to light sensitivity, wrap the microtubes containing novobiocin, tetracycline, and rifampicin antibiotics in aluminum foil. Antibiotics may be stored at −20 °C for extended periods. When preparing antibiotic solutions wear protective clothing, gloves, and eye/face protection.Disposable frozen mating starter cells can be stored at −70°C for extended periods. The use of donor and recipient cells at a concentration of approximately 5 × 10^8^ CFU mL^−1^ is mandatoryAlternatively, fresh donor and recipient cell cultures may be used.Donor cells (100 µL) and recipient cells (900 µL) are also processed separately with the same procedure and included as controls for the conjugation experiment. The 1/10 donor/recipient cell ratio is mandatory.Mating reactions can be frozen at –70 °C and plated later. Comparable numbers of transconjugants can be obtained from fresh or frozen mating reactions.Dilutions are routinely plated as follows: 10^−1^ and 10^−2^ of the conjugation mixture for transconjugants selection, 10^−6^ and 10^−7^ for donor cells counts, 10^−7^ and 10^−8^ for recipient cells counts.We constructed new *Streptococcus pneumoniae* strains, FP10 and FP11, to be used as standard conjugation recipients to transfer MGEs from the original encapsulated clinical isolates. These strains: (i) lack the capsule, (ii) contain a deletion in the *comC* gene for competence stimulating peptide (CSP) and are impaired in natural competence for genetic transformation, and (iii) harbor the *str-41* (FP10) and the *nov-1* (FP11) point mutations conferring resistance to streptomycin and novobiocin, respectively. The absence of the polysaccharide capsule on the bacterial surface increases the efficiency of ICEs conjugal transfer [9]. The impairment in natural competence for genetic transformation allows us to rule out the contribution of transformation to the genetic exchange during conjugation (F. Iannelli and G. Pozzi, unpublished). The availability of two strains with different resistance markers allows transconjugants selection and to transfer the genetic elements from transconjugants again. Plating of the mating reactions includes: (i) plates containing both antibiotics for the resistance marker of the donor genetic element and for the chromosomal resistance marker of recipient strain; (ii) plates containing antibiotic for the resistance marker of the genetic element of the donor strain; and (iii) plates containing antibiotic for the chromosomal resistance marker of recipient strain. Appropriate antibiotics are added to this 8 ml TSA layer at the following concentrations: 5 µg mL-1 chloramphenicol, 1 µg mL^−1^ erythromycin, 1000 µg mL^−1^ kanamycin, 10 µg mL^−1^ novobiocin, 400 µg mL^−1^ spectinomycin, 1000 µg mL^−1^ streptomycin, 5 µg mL^−1^ tetracycline, 25 µg mL^−1^ fusidic acid, 25 µg mL^−1^ rifampicin. The multilayer plating allows: (i) a slow diffusion of the antibiotics in the agar layer containing bacteria, (ii) the visualization of the colony’s three-dimensional structure, (iii) a better count of the colonies since the plate is transparent and colonies are generally larger than when spread on a plate, and (iv) the prevention of contact with ambient air favoring the growth of fastidious bacteria.Incubation can be extended to 48 h if required.At this stage, carefully check the phenotype of the colonies in order to exclude isolation of spontaneous mutants or colonies which might grow even in the absence of any genotype conferring resistance. We built a transilluminating box apparatus equipped with an inner white light illuminating an upper plexiglass cover. An overhead transparency plotted with petri dish-size grids overlays the plexiglass. Blood-agar plates can be adjusted over the grids so that each plate is divided into a total of 100 sectors [10]. Plates used for the genetic analysis of transconjugants include: (i) a plate containing both antibiotics for the resistance marker of the donor genetic element and for the chromosomal resistance marker of the recipient strain; (ii) a plate containing the antibiotic for the resistance marker of the donor genetic element; (iii) a plate containing the antibiotic for the chromosomal resistance marker of recipient strain; and (iv) a plate containing no antibiotics. In the absence of a transilluminator device, naked eye observation is possible, constructing a grid using a Petri dish lid.Confirm the phenotypes of transconjugants by PCR genotyping using primers for the amplification of the ICE-chromosome junctions and for ICE internal region such as the resistance genes.

## 5. Results

In this work, we report an established plate mating protocol for the conjugal transfer of large ICEs up to 81 kb in streptococci (*Streptococcus pneumoniae*, *Streptococcus agalactiae*, *Streptococcus gordonii*, *Streptococcus pyogenes*), *Enterococcus faecalis,* and *Bacillus subtilis* (Table 1 and Table 2). With this procedure, the transfer of the following genetic elements in the new *S. pneumoniae* recipients was obtained: (i) *S. pneumoniae* ICE Tn*5253* (size 65 kb) carrying the *cat* and *tet*(M) resistance genes [5,11,12], (ii) *S. pneumoniae* ICE Tn*5251* (size 18 kb) carrying the *tet*(M) [11], (iii) ICE*Sp*23FST81-like of *S. pneumoniae* type 23F genome strain ATCC 700669 (size 81 kb) (conjugation frequency 2.3 × 10^−6^ transconjugants per donor), and (iv) ICE Tn*5253*-like (size 81 kb) of *S. pneumoniae* type 6 genome strain 670–6B (conjugation frequency 2.7 × 10^−7^ transconjugants per donor). ICE Tn***5253*** was successfully transferred from the representative transconjugant FR58 to *S. pneumoniae* strains with different capsular types, *S. pyogenes*, *S. gordonii*, *S. agalactiae*, and transferred back from representative transconjugants of each bacterial species to *S. pneumoniae* (conjugation frequency varying from 4.4 × 10^−7^ to 6.7 × 10^−3^ to transconjugants per donor). ICE Tn*5251* is part of the composite *S. pneumoniae* ICE Tn*5253* and uses its highly efficient conjugation machinery to spread among bacterial strains. This conjugation protocol also allows the detection of rare events such as the autonomous transfer of Tn*5251*, as an independent ICE, from the *S. pneumoniae* host to *E. faecalis* [11]. Finally, we applied this protocol to transfer *S. pyogenes* ϕ1207.3 phage (size 53 kb, carrying the *mef* (A)-*msr* (D) macrolide resistance genes [13]), which moves through a mechanism resembling conjugation [14,15,16]. Conjugal transfer from the original 2812A clinical strain to the FP10 recipient occurred at a frequency of 3.8 × 10^−5^ transconjugants per donor and from the resulting transconjugant FR119 to *S. pyogenes* SF370 occurred with a frequency of 4.3 × 10^−6^ transconjugants per donor.

## 6. Concluding Remarks

In conclusion, the present conjugation protocol, based on plate mating, represents an efficient, low cost and easy-to-perform procedure to transfer large ICEs with a size up to 81 kb among streptococci and enterococci. This protocol allows compliance with much higher transfer frequencies and the transfer of elements that could not be moved using a classical filter mating protocol. Specifically, we obtained: (i) the autonomous transfer of ICE Tn*5251* element from *S. pneumoniae* to *E. faecalis* (<1.8 × 10^−8^ transconjugants per donor with the filter mating protocol [17]), and from here among different gram-positive species; (ii) the transfer of the 81 kb ICE*Sp*23FST81 element from the original clinical *S. pneumoniae* type 23 to pneumococcal conjugation recipients (<3.8 × 10^−8^ transconjugants per donor with the filter mating protocol); and (iii) *S. pyogenes* ϕ1207.3 phage lysogenic transfer to the *S. pneumoniae* FP10 recipient with a 5.4 fold increase compared to that obtained with the filter mating protocol.

## Figures and Tables

**Figure 1 mps-04-00059-f001:**
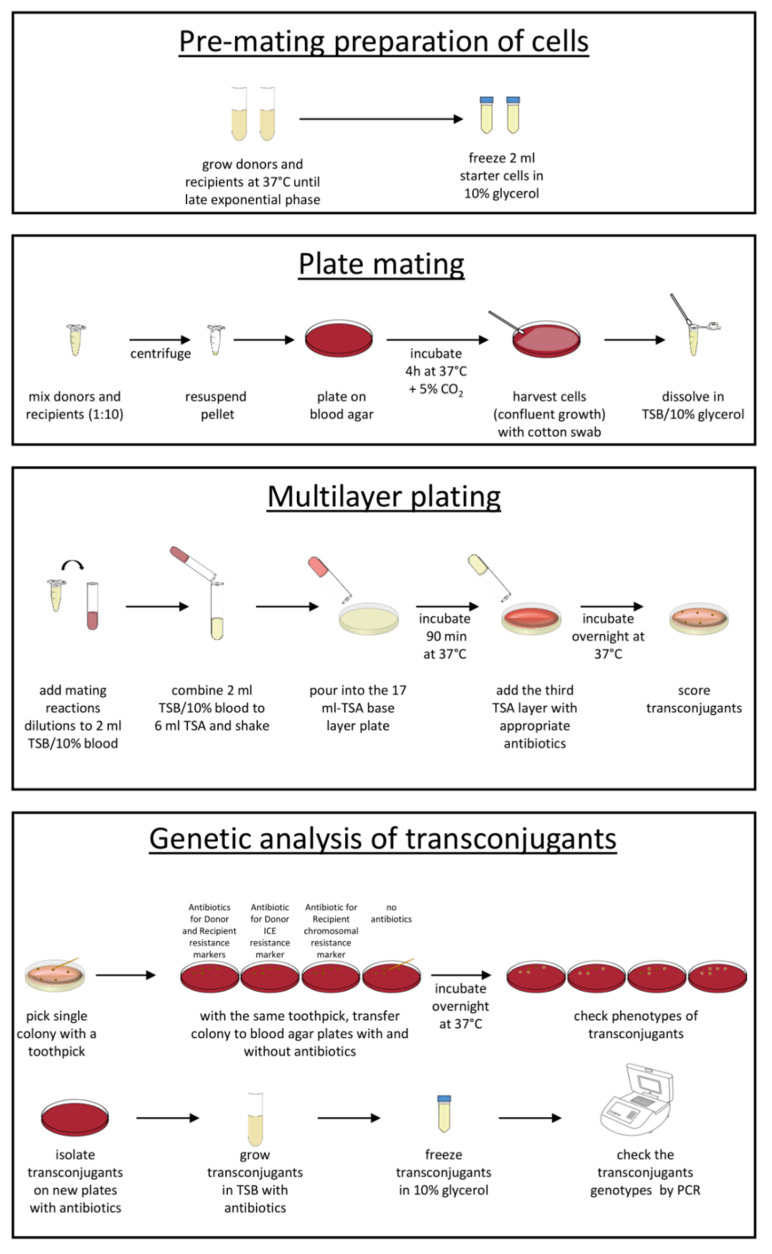
Schematic representation of the plate mating protocol for ICEs conjugal transfer in *streptococci* and *enterococci*.

**Table 1 mps-04-00059-t001:** ICEs conjugal transfer frequencies obtained with the plate mating protocol in streptococci and enterococci.

Donor Strain (Properties)	ICE (Size)	Recipient Strain (Properties)	Transfer Frequency ^a^	Genetic Analysis of Transconjugants(Primer Pairs) ^b^				Ref.
				*att*L–*att*Tn	*att*R–*att*Tn)	*tet*(M)	*cat*	
*S. pneumoniae* FR22 (FP10 transconjugant derivative, laboratory strain)	Tn*5253* (65-kb)	*S. pneumoniae* FP11 (rough laboratory strain)	1.6 × 10^−4^ ± 1.6 × 10^−5^	IF325-IF327	IF328-IF356	IF394-IF564	IF353-IF354	[11]
		*S. pneumoniae* FP47 (TIGR4 derivative, type 4 clinical strain)	3.6 × 10^−5^ ± 1.7 × 10^−6^	IF325-IF327	IF328-IF356	IF394-IF564	IF353-IF354	[11]
		*S. agalactiae* H36B (Ib clinical strain)	3.8 × 10^−6^ ± 5.3 × 10^−7^	IF560-IF327	IF328-IF561	IF394-IF564	IF353-IF354	[11]
		*S. gordonii* GP204 (V288 derivative, laboratory strain)	2.5 × 10^−5^ ± 5.5 × 10^−6^	IF512-IF327	IF328-IF513	IF394-IF564	IF353-IF354	[11]
		*S. pyogenes* SF370 (M1 clinical strain)	8.2 × 10^−6^ ± 7.5 × 10^−7^	IF509-IF327	IF328-IF510	IF394-IF564	IF353-IF354	[11]
		*E. faecalis* OG1RF (clinical strain)	5.4 × 10^−6^ ± 5.5 × 10^−7^	IF532-IF327	IF328-IF525	IF394-IF564	IF353-IF354	[11]
		*E. faecalis* JH2-2 (clinical strain)	8.6 × 10^−7^ ± 8.8 × 10^−8^	IF532-IF327	IF328-IF525	IF394-IF564	IF353-IF354	[11]
*S. pneumoniae* 670-6B (type 6 clinical strain)	Tn*5253*-like (81-kb)	*S. pneumoniae* FP11	2.7 × 10^−7^ ± 8.1 × 10^−8^	IF325-IF327	IF328-IF356	IF394-IF564	IF353-IF354	This study
*S. pneumoniae* ATCC700699 (type 23F clinical strain)	ICE*Sp*23FST81 (81-kb)	*S. pneumoniae* FP11	2.3 × 10^−6^ ± 1.0 × 10^−6^	IF345-IF855	IF327-IF347	IF394-IF564	IF353-IF354	This study
*S. pneumoniae* FR58 (FP11 transconjugant derivative)	Tn*5253* (65-kb)	*S. pneumoniae* FP58 (D39 derivative, type 2 clinical strain)	1.9 × 10^−5^ ± 4.6 × 10^−6^	IF325-IF327	IF328-IF356	IF394-IF564	IF354-IF353	[12]
		*S. pneumoniae* HB394 (A66 derivative, type 3 clinical strain)	4.4 × 10^−7^ ± 3.6 × 10^−8^	IF325-IF327	IF328-IF356	IF394-IF564	IF354-IF353	[12]
		*S. pneumoniae* FR55 (SP18-BS74 derivative, type 6 clinical strain)	1.3 × 10^−5^ ± 2.8 × 10^−6^	IF325-IF327	IF328-IF356	IF394-IF564	IF354-IF353	[12]
*S. agalactiae* FR67 (H36B transconjugant derivative)	Tn*5253* (65-kb)	*S. pneumoniae* FP11	1.1 × 10^−6^ ± 3.5 × 10^−7^	IF325-IF327	IF328-IF356	IF394-IF564	IF354-IF353	[12]
*S. gordonii* FR43 (GP204 transconjugant derivative)	Tn*5253* (65-kb)	*S. pneumoniae* FP11	8.3 × 10^−7^ ± 2.9 × 10^−7^	IF325-IF327	IF328-IF356	IF394-IF564	IF354-IF353	[12]
*S. pyogenes* FR40 (SF370 transconjugant derivative)	Tn*5253* (65-kb)	*S. pneumoniae* FP11	6.7 × 10^−3^ ± 1.0 × 10^−3^	IF325-IF327	IF328-IF356	IF394-IF564	IF354-IF353	[12]
*S. pneumoniae* FR22	Tn*5151* (18-kb)	*S. pneumoniae* FP47	1.5 × 10^−6^ ± 1.2 × 10^−8^			IF394-IF564		[11]
		*E. faecalis* OG1RF	5.4 × 10^−6^ ± 5.5 × 10^−7^			IF394-IF564		[11]
		*E. faecalis* JH2-2	8.6 × 10^−7^ ± 8.8 × 10^−8^			IF394-IF564		[11]
*S. gordonii* FR70 (GP204 transconjugant derivative)	Tn*5151* (18-kb)	S. pneumoniae FP11	3.1 × 10^−7^ ± 1.9 × 10^−7^			IF394-IF564		[11]
*S. pyogenes* FR71 (SF370 transconjugant derivative)	Tn*5151* (18-kb)	*S. pneumoniae* FP11	3.3 × 10^−5^ ± 1.3 × 10^−5^			IF394-IF564		[11]
*E. faecalis* FR64 (OG1RF transconjugant derivative)	Tn*5151* (18-kb)	*S. gordonii* GP204	4.8 × 10^−5^ ± 8.5 × 10^−6^			IF394-IF564		[11]
		*S. pyogenes* SF370	9.1 × 10^−7^ ± 2.8 × 10^−7^			IF394-IF564		[11]
		*E. faecalis* OG1SS (clinical strain)	1.3 × 10^−6^ ± 3.9 × 10^−7^			IF394-IF564		[11]
		*B. subtilis* 168 (laboratory strain)	1.6 × 10^−6^ ± 5.1 × 10^−7^			IF394-IF564		[11]
*S. pneumoniae* FR73 (FP47 transconjugant derivative)	Tn*5151* (18-kb)	*S. pneumoniae* FP22 (rough D39 derivative )	2.6 × 10^−8^ ± 8.9 × 10^−9^			IF394-IF564		[11]
		*S. pneumoniae* FP23 (rough TIGR4 derivative)	1.2 × 10^−7^ ± 6.1 × 10^−8^			IF394-IF564		[11]
*S. pyogenes* 2812A (clinical strain)	Φ1207.3 (53-kb)	*S. pneumoniae* FP10	3.8 × 10^−5^ ± 7.6 × 10^−6^	IF281-IF127	IF162-IF282			This study
*S. pneumoniae* FR119 (FP10 transconjugant derivative)	Φ1207.3 (53-kb)	*S. pyogenes* SR300 (SF370 derivative strain)	4.3 × 10^−6^ ± 1.1 × 10^−6^	IF302-IF127	IF162-IF303			This study

^a^ Conjugation frequency is expressed as CFU of transconjugants per CFU of donor. The results are presented as the mean of at least 3 mating experiments. ^b^ Transconjugants are selected for acquisition of antibiotic resistance and their genotype characterized by PCR. The presence of the *att*L–*att*Tn junction, *att*R–*att*Tn junction, *tet*(M) and *cat* was investigated in the transconjugants carrying Tn*5253*-family elements. Due to the unspecific integration of Tn*5251* into the bacterial chromosome, the presence of *tet*(M) was investigated in the transconjugants carrying the element. Primers sequences are reported in Table 2.

**Table 2 mps-04-00059-t002:** PCR oligonucleotide primers for transconjugants genotyping.

Name	Sequence (5′ to 3′)	Target (Bacterial Species)	Direction
IF327 IF328	CAATATAGCGTGATGATTGTAAT AGTGAGAATCAAATCAGAGGTT	5’ end of Tn*5253*, Tn*5253*-like 3’ end of Tn*5253*, Tn*5253*-like	Reverse Forward
IF325 IF356	ACAAGAACTGTTTGGACATCAT GACTAGATAGAGGCAAGCGT	Tn*5253*, Tn*5253*-like chromosomal integration site (*S. pneumoniae*)	Forward Reverse
IF560 IF561	AACGAAACCTATCAGCGGAA TTTGGGTTTGTCTCCGACGA	Tn*5253*, Tn*5253*-like chromosomal integration site (*S. agalactiae*)	Forward Reverse
IF512 IF513	TGCTTTAGGAGATGTTGAGTT ACCGCAGACTGTTCTTTAGA	Tn*5253*, Tn*5253*-like chromosomal integration site (*S. gordonii*)	Forward Reverse
IF509 IF510	AAGTAGAAATGGCGAAGTGAA GACTAGAAAGTGGTAAGCGT	Tn*5253*, Tn*5253*-like chromosomal integration site (*S. pyogenes*)	Forward Reverse
IF532 IF525	GCCTATGGGATTGCTACACC GGTTACGGGAAGAAAGCGGT	Tn*5253*, Tn*5253*-like chromosomal integration site (*E. faecalis*)	Forward Reverse
IF855 IF327	ACCAAATTCCTGCCAGAGTTGA CAATATAGCGTGATGATTGTAAT	5’ end of ICE*Sp*23FST81 3’ end of ICE*Sp*23FST81	Reverse Forward
IF345 IF347	ATGGTAATCATCTAAAAATGTCAC CACCAGCACTTGTTAAAGAAG	ICE*Sp*23FST81chromosomal integration site (*S. pneumoniae*)	Forward Reverse
IF394 IF564	GCTATAGTATAAGCCATACTT GAAGTGACTTGTGCTCTGCT	Tn*5253*-family *tet* (M) resistance gene	Forward Reverse
IF354 IF353	CATTCTCTGGTATTTGGACTC CTCTCCGTCGCTATTGTAAC	Tn*5253*-family *cat* resistance gene	Forward Reverse
IF127 IF162	TGTTCTTCATCTACTACGACTG TGATGATTATATAAATTGTGAGTT	5’ end of Φ1207.3 3’ end of Φ1207.3	Reverse Forward
IF281 IF282	AGGTGGTAAGGCAGAATC GCACCTTTGTTTGAGTCG	Φ1207.3 chromosomal integration site (*S. pneumoniae*)	Forward Reverse
IF302 IF303	AATAGATGTAGGTGGGCG AGCTTTGGCAACCACTTC	Φ1207.3 chromosomal integration site (*S. pyogenes*)	Forward Reverse

## Data Availability

Not applicable.

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
