# Peer review of "A Mating Procedure for Genetic Transfer of Integrative and Conjugative Elements (ICEs) of Streptococci and Enterococci"

_mps, 2021, doi:10.3390/mps4030059_

Round 1

Reviewer 1 Report

A paper „A mating procedure for genetic transfer of integrative and conjugative elements (ICEs) of streptococci and enterococci” by Ianelli and co-workers provides a detailed and easy to follow description for a mating procedure to be used for several species of Gram-positive bacteria. While such paper certainly will be useful for several researches in the field, I have some points to address:

  1. Part 3.5. does this system allow to manipulate and count colonies using only a naked eye or the transilluminator device is indispensable?
  2. Page 5. It would be easier for the reader if the data on conjugation (and conjugation efficiencies, with e.g. standard deviations as a measure of protocol reproducibility) were included in a more systematic manner as a Table, where strain names and provenance (clinical or laboratory) and an MGE name/size should be mentioned.
  3. Authors may add a brief paragraph at the end of paper, indicating what the advances of their protocol compared to previously known approaches are.

Author Response

Response to Reviewer 1 Comments

Point 1:

Part 3.5. does this system allow to manipulate and count colonies using only a naked eye or the transilluminator device is indispensable?

Response 1:

THE PROTOCOL ALLOWS TO COUNT CFUs USING A NAKED EYE, THE TRANSILLUMINATING BOX-APPARATUS IS NOT INDISPENSABLE, A CLARIFICATION SENTENCE WAS ADDED IN THE REVISED PAPER (Section Notes: note 10, lines 194-195)

Point 2:

Page 5. It would be easier for the reader if the data on conjugation (and conjugation efficiencies, with e.g. standard deviations as a measure of protocol reproducibility) were included in a more systematic manner as a Table, where strain names and provenance (clinical or laboratory) and an MGE name/size should be mentioned.

Response 2:

TABLE 1, CONTAINING THE CONJUGATION EFFICIENCIES WITH STANDARD DEVIATIONS, STRAINS CHARACTERISTICS AND OTHER INFORMATION, WAS INCLUDED IN THE REVISED PAPER

Point 3:

Authors may add a brief paragraph at the end of paper, indicating what the advances of their protocol compared to previously known approaches are.

Response 3:

THE ADVANCES OF THIS PROTOCOL COMPARED TO THE FILTER MATING PROTOCOL WERE DESCRIBED IN THE REVISED PAPER (Section Concluding remarks, lines 226-233)

Reviewer 2 Report

Horizontal gene transfer mechanism, such as integrative and conjugative elements (ICEs), contributes to bacterial adaptation to changing environments due to the acquisition of virulence factors, antibiotic resistance genes, and plays an important role in prokaryotic evolution. Francesco et’ al described a simple protocol for conjugal transfer of large ICEs in streptococci, Enterococcus faecalis and Bacillus subtilis. Notably, this protocol allows the transfer of large ICEs with a size up to 81-kb. And interestingly, this protocol can be applied to generate conjugation in a Streptococcus pneumoniae strain without natural competence. However, this protocol is too basic and a few points shall be improved to quantify the publication standard of MPs.

  1. It is suggested to add a diagram of this protocol to help readers to follow.
  2. The authors claimed that this protocol could be applied in multiple bacterial species, including streptococci, Enterococcus faecalis and Bacillus subtilis. However, relevant data was not showed in the manuscript. An “Expected Results” section with detailed analyzed data is needed to illustrate that this protocol is practical.
  3. What are the factors affecting horizontal gene transfer efficiency, the size of ICE, the OD of the starters, or the ratio of donor/recipients? The authors need to display how they optimize the steps.
  4. Since the authors performed in vitro conjugation using a S. pneumonia strain without competence capability, it is needed to discuss the mediators of this process in the discussion section. Is it achieved through Type IV Secretion Systems? What is the conjugation mechanism in Enterococcus and Bacillus?

Author Response

Response to Reviewer 2 Comments

Point 1:

  1. It is suggested to add a diagram of this protocol to help readers to follow.

Response 1:

FIGURE 1, CONTAINING A DIAGRAM OF THE PROTOCOL, WAS ADDED IN THE REVISED PAPER AS SUGGESTED

Point 2:

  1. The authors claimed that this protocol could be applied in multiple bacterial species, including streptococci, Enterococcus faecalis and Bacillus subtilis. However, relevant data was not showed in the manuscript. An “Expected Results” section with detailed analyzed data is needed to illustrate that this protocol is practical.

Response 2:

TABLE 1, CONTAINING THE RESULTS OBTAINED WITH THIS PROTOCOL, WAS ADDED IN THE REVISED PAPER

Point 3:

  1. What are the factors affecting horizontal gene transfer efficiency, the size of ICE, the OD of the starters, or the ratio of donor/recipients? The authors need to display how they optimize the steps.

Response 3:

DURING OPTIMIZATION OF THE PROTOCOL DONOR AND RECIPIENT CELLS WERE MIXED AT DIFFERENT RATIOS AND OPTICAL DENSITY AND WE ESTABILSHED THAT TRANSFER EFFICIENCY IS STRONGLY INFLUENCED BY BOTH PARAMETERS. THE OPTIMAL PARAMETERS ARE 1:10 DONOR/RECIPIENT RATIO AND AN OPTICAL DENSITY OF 0.8 (corresponding to approximately 5x108 CFU mL-1). TWO CLARIFICATION SENTENCES WERE ADDED IN THE REVISED PAPER (Section Notes: note 3 (line  139-140) AND note 5 (line 143-144)

Point 4:

  1. Since the authors performed in vitro conjugation using a S. pneumonia strain without competence capability, it is needed to discuss the mediators of this process in the discussion section. Is it achieved through Type IV Secretion Systems? What is the conjugation mechanism in Enterococcus and Bacillus?

Response 4:

WE DEVELOPED A PROTOCOL FOR CONJUGAL TRANSFER AT HIGH EFFICIENCY OF ICEs IN GRAM POSITIVE BACTERIA WHICH REPRESENTS A USEFUL TOOL FOR ICEs FUNCTIONAL STUDIES. ALL THE TRANSFERRED ELEMENTS POSSESS A TYPE IV SECRETION SYSTEM, WHICH IS PROBABLY RESPONSIBLE FOR THE CONJUGAL TRANSFER. WE HAVE EVIDENCE THAT DELETION OF THE TN5253 PUTATIVE TYPE IV SECRETION SYSTEM ABOLISHES CONJUGAL TRANSFER (UNPUBLISHED DATA). 

Reviewer 3 Report

The authors described a mating procedure that allows ICEs transfer between streptococcal and enterococcal strains. They used S. pneumonia FP10 and FP11 as standard recipient strains. This protocol was shown to be efficient to detect the transfer of a large number of ICEs. The development of robust and efficient mating procedure is needed to detect the transfer and to explore at functional level the biology of these mobile genetic elements.

Here are some points to be discussed:

  • The protocol is well described but we believe that a graphical protocol describing the different steps will improve the readers’ understanding.
  • A list of the donor and recipient strains including ICEs names and their transfer frequencies would make the results much easier to understand.
  • A comparison of the transfer frequencies of the different ICEs tested using this new procedure with the classical filter mating conjugation protocol is required to highlight the advantages of the described protocol compared to the classical one.
  • The usage of the multilayer plating is time consuming. It will be helpful to describe the advantages of this procedure compared to the classical plating.
  • Line 105: The dilution step is critical for transconjugants detection. a detailed description of the “appropriate dilution” of each donor-recipient couple is required.
  • Line 136: remove “of” from “of and “.
  • Line 147: the role of the envelope components on conjugative transfer especially exopolysaccharide was described in the literature. The authors can cite the paper by Dahmane et al 2018 1128/AEM.02109-17 .
  • Genetic analysis (PCR amplifications) of transconjugants must be showed for some ICEs including their appropriate controls. Since these are important aspects of this mating protocol, a description of the analyzed region of each ICE and the sequences of the used primers must be listed in a table.

Author Response

Response to Reviewer 3 Comments

Point 1:

  • The protocol is well described but we believe that a graphical protocol describing the different steps will improve the readers’ understanding.

Response 1:

FIGURE 1, CONTAINING A DIAGRAM OF THE PROTOCOL, WAS ADDED IN THE REVISED VERSION AS SUGGESTED

Point 2:

  • A list of the donor and recipient strains including ICEs names and their transfer frequencies would make the results much easier to understand.

Response 2:

TABLE 1, CONTAINING THE CONJUGATION EFFECIENCIES, STRAINS CHARACTERISTICS AND OTHER INFORMATION, WAS ADDED IN THE REVISED PAPER

Point 3:

  • A comparison of the transfer frequencies of the different ICEs tested using this new procedure with the classical filter mating conjugation protocol is required to highlight the advantages of the described protocol compared to the classical one.

Response 3:

CLARIFICATION SENTENCES WITH A COMPARISON OF TRANSFER FREQUENCIES OBTAINED WITH THE PLATE MATING PROTOCOL AND THE FILTER MATING PROTOCOL WERE ADDED IN THE REVISED PAPER (Section Concluding remarks, lines 226-233)

Point 4:

  • The usage of the multilayer plating is time consuming. It will be helpful to describe the advantages of this procedure compared to the classical plating.

Response 4:

THE MULTILAYER PLATING ALLOWS: i) A SLOW DIFFUSION OF THE ANTIBIOTICS IN THE AGAR LAYER CONTAINING BACTERIA, ii) THE VISUALIZATION OF THE COLONY THREE-DIMENSIONAL STRUCTURE, iii) A BETTER COUNT OF THE COLONIES SINCE THE PLATE IS TRANSPARENT AND COLONIES ARE GENERALLY LARGER THAN WHEN SPREAD ON A PLATE, and iv) PREVENTS CONTACT WITH AMBIENT AIR FAVOURING GROWTH OF FASTIDIOUS BACTERIA. A CLARIFICATION SENTENCE DESCRIBING THE ADVANTAGES WAS ADDED IN THE REVISED PAPER (Section Notes: note 8, lines 170-175)

Point 5:

  • Line 105: The dilution step is critical for transconjugants detection. a detailed description of the “appropriate dilution” of each donor-recipient couple is required.

Response 5:

A CLARIFICATION SENTENCE DESCRIBING THE DILUTION DETAILS WAS ADDED IN THE REVISED PAPER (Section Notes: note 7, lines 147-149)

Point 6:

  • Line 136: remove “of” from “of and “.

Response 6:

CORRECTED

Point 7:

  • Line 147: the role of the envelope components on conjugative transfer especially exopolysaccharide was described in the literature. The authors can cite the paper by Dahmane et al 2018 1128/AEM.02109-17 .

Response 7:

THE CITATION SUGGESTED WAS ADDED IN THE REVISED PAPER

Point 8:

  • Genetic analysis (PCR amplifications) of transconjugants must be showed for some ICEs including their appropriate controls. Since these are important aspects of this mating protocol, a description of the analyzed region of each ICE and the sequences of the used primers must be listed in a table.

Response 8:

TABLE 1, CONTAINING THE PRIMERS USED AND THE REGION AMPLIFIED DURING GENETIC ANALYSIS, WAS ADDED IN THE REVISED PAPER

Round 2

Reviewer 2 Report

The authors have revised the manuscript according to previous suggsetions.